# Effects of transcranial magnetic stimulation over the left posterior superior temporal gyrus on picture-word interference

**Vitória Piai[1,2]***, **Laura Nieberlein[3]**, **Gesa Hartwigsen****[3]***

1 Donders Centre for Cognition, Radboud University, Nijmegen, the Netherlands, 2 Department of Medical Psychology, Donders Centre for Medical Neuroscience, Radboudumc, Nijmegen, the Netherlands, 3 Lise Meitner Research Group Cognition and Plasticity, Max Planck Institute for Human Cognitive and Brain Sciences, Leipzig, Germany

* v.piai@donders.ru.nl (VP); hartwigsen@cbs.mpg.de (GH)

**Data Availability Statement:** The data and materials fare available at the open science framework: https://tinyurl.com/y5vky8y3.

## Abstract

Word-production theories argue that during language production, a concept activates multiple lexical candidates in left temporal cortex, and the intended word is selected from this set. Evidence for theories on spoken-word production comes, for example, from the picture-word interference task, where participants name pictures superimposed by congruent (e.g., picture: rabbit, distractor "rabbit"), categorically related (e.g., distractor "sheep"), or unrelated (e.g., distractor "fork") words. Typically, whereas congruent distractors facilitate naming, related distractors slow down picture naming relative to unrelated distractors, resulting in semantic interference. However, the neural correlates of semantic interference are debated. Previous neuroimaging studies have shown that the left mid-to-posterior STG (pSTG) is involved in the interference associated with semantically related distractors. To probe the functional relevance of this area, we targeted the left pSTG with focal repetitive transcranial magnetic stimulation (rTMS) while subjects performed a picture-word interference task. Unexpectedly, pSTG stimulation did not affect the semantic interference effect but selectively increased the congruency effect (i.e., faster naming with congruent distractors). The facilitatory TMS effect selectively occurred in the more difficult list with an overall lower name agreement. Our study adds new evidence to the causal role of the left pSTG in the interaction between picture and distractor representations or processing streams, only partly supporting previous neuroimaging studies. Moreover, the observed unexpected condition-specific facilitatory rTMS effect argues for an interaction of the task- or stimulus-induced brain state with the modulatory TMS effect. These issues should be systematically addressed in future rTMS studies on language production.

## 1 Introduction

Successful human communication requires speakers to retrieve information from long-term memory quickly and accurately. According to word-production theories, a concept, for example denoted by a picture, activates multiple lexical candidates (in left middle temporal cortex),

**Funding:** GH is supported by the German Research Foundation (DFG, HA-6414/3-1 & HA-6314/4-1) and by the Max Planck Society. VP is supported by the Netherlands Organisation for Scientific Research (grant numbers 451-17-003 and Gravitation Grant 024.001.006 to the Language in Interaction Consortium) The funders had no role in study design, data collection and analysis, decision to publish, or preparation of the manuscript.

**Competing interests:** The authors have declared that no competing interests exist.

and the intended word is selected from this set [1–3]. Activation spreads through the different levels, from concepts to lexical items ("lemmas"), phonemes, and motor programs.

The picture-word interference paradigm has been central to investigating stages of spoken word production. Participants name pictures with superimposed distractor words. The picture primes the distractor and the distractor primes the picture name via conceptual connections in memory [4]. Accordingly, effects emerging from the interaction between properties of the distractor and the picture have been important for theorizing about different stages of production [5–7]. Categorically related distractors (e.g., picture: rabbit, distractor "sheep") typically slow down naming relative to unrelated distractors (e.g., "fork"), which is termed semantic interference. In contrast, congruent distractors (e.g., "rabbit") speed up naming relative to related and unrelated distractors due to convergence at the conceptual, lexical, and phonological levels [for review see 8]. Recently, the locus of the semantic interference effect has been the topic of much debate in the field [e.g. 9, 10], despite the well-established evidence for its existence [11].

The neural correlates of this effect have also remained largely unclear [12]. This is certainly the case for different structures in the left temporal lobe, an area thought to play a major role in various stages of production [13]. Involvement of mid-to-posterior superior temporal (STG) and middle temporal (MTG) gyri has been reported, with effects sometimes going in opposite directions [14–17]. Patients with lesions in the temporal lobe, overlapping in the mid-to-posterior MTG, may sometimes show increased semantic interference effects [18, 19]. In a neuroimaging study focusing on control demands across verbal and non-verbal tasks, cross-domain activity was found in the mid-to-posterior STG for the most difficult conditions [for picture-word interference, the categorically related condition; 20]. Two previous neuroimaging studies observed the same area (with very similar coordinates) in association with the semantic interference effect [14, 16]. However, the causal role of this area in semantic interference from categorically related distractors is currently unknown.

To probe the functional relevance of the left mid-to-posterior STG (pSTG henceforth) in semantic interference, we targeted this area with focal repetitive transcranial magnetic stimulation (rTMS) while healthy participants named pictures paired with congruent, categorically related, or unrelated distractors. An important advantage of focal TMS-induced perturbations relative to studies of structural lesions is that they are not confounded by chronic processes mediating functional recovery [21, 22]. Note that although the employed high-frequency TMS approach (short bursts at 10 Hz) is usually expected to increase neuronal excitability in the motor system, numerous studies of cognition have reported behavioral disruption with this protocol [e.g., 23 for review]. Specifically, 10 Hz rTMS has been demonstrated to transiently impair different language processes when applied during task execution, leading to increased response latencies or decreased task accuracy in a number of previous language comprehension studies [e.g., 24–27]. Consequently, we expected that focal perturbation with 10 Hz rTMS of the left pSTG region, identified in previous neuroimaging studies on semantic interference [14, 20], should interfere with word production stages, resulting in stronger semantic interference (i.e., related vs. unrelated distractors) for pSTG stimulation relative to the stimulation of a control site not involved in spoken-word planning stages.

## 2 Methods

### 2.1 Materials

Two lists of materials were designed and pretested (details below in Section 2.3.2). Each list contained 58 pictures from 14 semantic categories. Each category was represented with 3 to 5 pictures, with an equal number of pictures per category in both lists. The picture names were high-frequency nouns in German. Pictures were coloured photographs on a white background

**Fig 1. Overview of the experiment.** A. Example of the three distractor conditions for the picture of a rabbit. Distractors were in the response set, that is, the distractors "fork" and "sheep" also appeared as pictures in the experiment with other distractor words. B. Example of two experimental trials. Repetitive transcranial magnetic stimulation (five pulses at 10 Hz) was applied with picture onset. C. Stimulated sites. pSTG = mid-to-posterior superior temporal gyrus.

[28; and from the internet]. German distractors were derived from the picture names, i.e., distractors were in the response set, yielding semantically related distractors (e.g., picture: rabbit, distractor "sheep"), unrelated distractors (e.g., picture: rabbit, distractor "fork"), and congruent distractors (e.g., picture: rabbit, distractor "rabbit"). The related and unrelated distractors were phonologically and orthographically unrelated to the picture. Distractors were presented superimposed onto the pictures in a centered position (see Fig 1A for an example).

The two stimulus lists were matched as much as possible for length (in number of syllables), lemma frequency, phonological familiarity, phonological regularity, name agreement, and onset phoneme category (in terms of voicing and manner of articulation). Lemma frequency, phonological familiarity, and phonological regularity for the picture names (and distractors) of both stimulus lists were calculated with dlexDB [http://dlexdb.de/; 29]. The Java application DISCO20 by "linguatools" (©linguatools, 2007–2018) was used to operationalise the semantic relationship between picture names and distractors in the related and unrelated conditions. Name agreement (percentage of individuals that named a given picture by its target name) and H-index [30] were obtained by asking a different group of 15 German native speakers to name all pictures with the first word that came to mind. The H-index for each picture was calculated

as

$$H = \sum_{i=1}^{k} p_i log_2 \left(\frac{1}{p_i}\right)$$

where k refers to the number of different names given to a picture and $p_i$ to the proportion of participants that gave each one of the names. The H-index captures not only the percentage of name agreement across individuals, but also the distribution of names across those individuals [30]. Larger H values indicate not only less name agreement, but also more variability in the names given to the same picture. The test was administered online and participants typed their responses. Pictures were presented in a unique randomised order for each participant.

## 2.2 Pretest of the materials

Given that the TMS experiment consisted of two sessions (pSTG and control-site stimulation), a behavioral experiment without TMS was first carried out to pretest the materials divided over two stimulus lists. We expected to replicate the previously described semantic interference and congruency facilitation effects in this experiment.

**2.2.1 Participants.** The pretest was conducted with ten native monolingual German speakers (five females) aged 20–33 years. All participants were right-handed [Edinburgh Handedness Inventory, 31; laterality index > 80%] and recruited via the local subject database. None of the participants had a history of psychiatric or neurological problems. Each participant received a compensation of €8 per hour. Prior to the start of the experiment, participants signed a written informed consent. The study was conducted in accordance with the guidelines of the Declaration of Helsinki and approved by the local ethics committee.

**2.2.2 Procedure.** The pretest consisted of two blocks (list 1 and 2, counterbalanced), corresponding to the two TMS sessions in the experiment proper. A break of 10 minutes separated the two blocks. Before each block, participants were familiarised with the items of the list by viewing all images with their correct names. Then, a short practice with 12 trials followed. Practice items were not part of the experimental lists. Each picture was repeated three times within a session, once with each distractor type. Unique experimental lists were generated from the two main lists, yielding one unique pseudorandomised and balanced list of each main list for each participant. The same condition was repeated at most three times in succession and at least ten different pictures intervened between repetitions of the same picture. The software Presentation (Neurobehavioral Systems, Inc., Albany, Calif., USA) controlled stimulus presentation and the recording of the responses. Each trial began with a fixation cross on the screen for 2 sec. Picture-word stimuli were then displayed for 2 sec. Participants were instructed to name the pictures as quickly and correctly as possible, ignoring the distractor words.

**2.2.3 Analysis.** Response times (RTs) were marked offline relative to picture onset using Praat [32] by LN, blinded for the condition a picture-naming trial belonged to. Incorrect responses were marked and excluded from the RT analyses, as well as responses slower than 3 s. Single-trial RT and accuracy were analysed with linear and logistic mixed-effects models, respectively, given that RTs are a continuous variable, which can be analysed with a linear regression since it meets the model's assumption of a continuous outcome, whereas accuracy is a binary variable (correct or incorrect) and, therefore, should not be analysed with a linear regression. Models were fitted with the lme4-package [33]. The RT model had the following structure: as fixed effects, we used distractor condition, list, block, the pictures' name agreement, frequency, and the interaction between list and condition, block and list, and name agreement and list [given the evidence that name agreement is the most important predictor

of naming latencies, 34]. By-participant and by-item random intercepts were used. Condition, list, and block were entered as repeated contrasts [35; MASS package, 36]. The error model using the same structure failed to converge. Therefore, we reduced the model's complexity in the fixed-effects structure, including only condition, list, and their interaction. Significance of effects was obtained using the Satterthwaite approximation [lmerTest-package, 37].

**2.2.4 Results.** The group-averaged RTs and error rates for each distractor condition and stimulus list are shown in Table 1. Details of the statistical analyses are shown in Table 2. As expected, in the RTs, a congruency effect was found (i.e., responses were slower for unrelated than for congruent trials, $p < 0.001$), as well as semantic interference (i.e., responses were slower for related than for unrelated trials, $p < 0.001$). A significant difference was observed between the two stimulus lists ($p = 0.004$), such that pictures in list 1 were named faster than in list 2. There was a name agreement effect in list 2 (i.e., the lower the name agreement, the slower the responses were, $p < 0.001$), but not in list 1 ($p = 0.761$). No interaction was found between distractor conditions and list ($p$s > 0.627). In the errors, a congruency effect was found (i.e., more errors for unrelated than for congruent trials, $p < 0.001$), as well as semantic interference (i.e., more errors for related than for unrelated trials, $p = 0.033$).

Thus, the expected distractor effects (i.e., congruent and related conditions differ from the unrelated condition) were found in both lists. Despite all the matching at the level of the lists, list 2 turned out to be more difficult than list 1, but this difficulty difference did not interact with distractor condition. Given the name agreement effect for list 2, it is likely that name agreement played a more important role in slowing down naming in list 2 than in list 1. The effect of name agreement can emerge at an early level (e.g., the picture is ambiguous with respect to the object it depicts) or later, i.e., the depicted object has several names [e.g., 38]. Accordingly, our name-agreement effects likely have loci at the semantic, lexical, and phonological stages [e.g., 39–41]. For the related and unrelated conditions, name agreement would affect only the picture. By contrast, for the congruent condition, lower name agreement translates into less congruency between picture name and distractor, and thus less facilitation from their converging semantic, lexical, and phonological levels.

## 2.3 TMS Experiment

**2.3.1 Participants.** Twenty-four native monolingual German speakers (12 females), aged 21–35 years (mean = 27.17, SD = 3.749), none of which took part in the pretest or name-agreement experiment, participated in our study. All participants were right-handed [Edinburgh Handedness Inventory, 31; mean laterality quotient = 91.42, SD = 9.83] with normal or corrected-to-normal vision. None of the participants reported any psychiatric and neurological disturbances or contraindications for TMS. Participants received €10/hour compensation. Prior to the experiment, participants signed an informed consent form and completed a questionnaire to rule out any risk factors. The study was conducted in accordance with the Declaration of Helsinki and approved by the local ethics committee.

**Table 1. Results of the pretest.**

| Distractor condition | List 1 RT (SD) | List 2 RT (SD) | List 1 EP% (SD) | List 2 EP% (SD) |
|---|---|---|---|---|
| Congruent | 724 (115) | 766 (130) | 0.172 (0.545) | 0.690 (1.206) |
| Unrelated | 844 (89) | 889 (125) | 2.931 (1.999) | 3.966 (2.936) |
| Related | 869 (98) | 922 (125) | 4.828 (5.440) | 5.517 (2.792) |

Group mean response time (RT, in ms), error percentage (EP%), and standard deviation (SD) for each distractor condition and stimulus list.

**Table 2. Inferential statistics for the response times and errors of the pretest.**

| Response Times | | | | |
|---|---|---|---|---|
| Condition/parameter | B | SE | t | p |
| Intercept | 897.43 | 39.30 | 22.84 | <0.001 |
| Unrelated vs related | -29.51 | 7.55 | -3.91 | <0.001 |
| Congruent vs unrelated | -122.56 | 7.46 | -16.43 | <0.001 |
| List 2 vs list 1 | 135.91 | 46.71 | 2.91 | 0.004 |
| Block 2 vs block 1 | -9.05 | 6.12 | -1.48 | 0.139 |
| Frequency | -0.23 | 0.12 | -1.88 | 0.060 |
| Unrelated vs related: list 2 vs list 1 | -7.34 | 15.10 | -0.49 | 0.627 |
| Congruent vs unrelated: list 2 vs list 1 | -4.74 | 14.92 | -0.32 | 0.751 |
| List 2 vs list 1: session 2 vs session 1 | 195.98 | 127.02 | 1.54 | 0.123 |
| List [1]: Name agreement | -0.12 | 0.40 | -0.30 | 0.761 |
| List [2]: Name agreement | -1.20 | 0.33 | -3.61 | <0.001 |
| Errors | | | | |
| Condition/parameter | Odds ratios | SE | z | p |
| Intercept | 105.36 | 0.34 | 13.72 | <0.001 |
| Unrelated vs related | 1.59 | 0.22 | 2.14 | 0.033 |
| Congruent vs unrelated | 10.87 | 0.59 | 4.07 | <0.001 |
| List 2 vs list 1 | 0.54 | 0.46 | -1.33 | 0.184 |
| Unrelated vs related: list 2 vs list 1 | 0.84 | 0.44 | -0.41 | 0.680 |
| Congruent vs unrelated: list 2 vs list 1 | 0.34 | 1.17 | -0.91 | 0.362 |

B = beta coefficient; SE = standard error.

**2.3.2 TMS-materials: Final set.** For the TMS study, the controlled German materials of the pretest were widely adopted. Two images that were often misrecognised in the pretest (Section 2.1) were replaced by clearer images ("TV" and "pen"). One four-syllable word ("Klarinette") was replaced by a two-syllable word ("Orgel"). Details of this set of materials are shown in Table 3. Comparisons between the two lists, and between related and unrelated items when relevant, were made with independent samples t-tests and the degrees of freedom were corrected where needed with the Welch modification.

No systematic differences between the two lists were found for length, lemma frequency, phonological familiarity, phonological regularity, name agreement nor in H-index (all $p$s > 0.103, see Table 3). Within each list, a significant difference was found in semantic similarity between the picture names and the distractors as a function of distractor type ($p$s <0.001), which is to be expected due to higher semantic similarity between picture and distractor for the related vs unrelated conditions. Comparing the two lists, no systematic differences were found in the semantic relationship between the picture names and the distractors ($p$s > 0.384). Onset phonemes of picture names were, for list 1 and 2 respectively, nasal: 4 vs 1; voiced approximant: 1 vs 4; voiced fricative: 1 vs 0; voiced plosive: 13 vs 8; voiceless affricate: 3 vs 3; voiceless fricative: 16 vs 17; voiceless plosive: 13 vs 16; vowel: 6 vs 5; grapheme "r" (voiced uvular fricative or trill, depending on the regiolect): 1 vs 4. A chi-squared test indicated no differences in distributions between the two lists ($X^2(8) = 8.55$, $p = 0.381$).

**2.3.3 Design and procedure.** The study followed a 2x3x2 within-subject design including the factors stimulation site (pSTG, vertex), distractor condition (congruent, unrelated, related) and list (1, 2). Stimulation sites were targeted in different sessions with an inter-session interval of at least seven days (mean = 12.08, SD = 12.18), with the site order counterbalanced across

**Table 3. Properties of the materials in the two lists of the TMS experiment and their comparison by means of independent samples t-tests (degrees of freedom are corrected where needed with the Welch modification).**

|  | List 1 | List 2 |  |
| --- | --- | --- | --- |
| *Parameter* | Median (SD) | Median (SD) | Comparison |
| Length in syllables | 2 (0.683) | 2 (0.700) | t(114) = -0.403, p = 0.688, 95%CI [-0.306, 0.203] |
| Frequency | 9.66 (37.74) | 8.00 (54.93) | t(101) = 0.295, p = 0.768, 95%CI [-14.777, 19.946] |
| Familiarity | 25.02 (177.83) | 23.85 (194.13) | t(114) = -0.0200, p = 0.984, 95%CI [-69.166, 67.795] |
| Regularity | 11.56 (172.45) | 12.20 (100.59) | t(92) = 0.689, p = 0.493, 95%CI [-34.008, 70.129] |
| Name agreement | 100 (20.03) | 93.33 (25.69) | t(114) = -1.122, p = 0.264, 95%CI [-0.278, 0.077] |
| H-index | 0 (0.517) | 0.42 (0.447) | t(114) = 1.639, p = 0.104, 95%CI [-1.461,15.484] |
| *Semantic similarity per condition* |  |  |  |
| Related | 0.484 (0.166) | 0.439 (0.154) | t(114) = 0.871, p = 0.385, 95% CI [-0.033, 0.085] |
| Unrelated | 0.158 (0.094) | 0.127 (0.094) | t(114) = 0.270, p = 0.788, 95% CI [-0.030, 0.039] |

SD = standard deviation; CI = confidence interval.

participants. Stimulation site, session, and list were administered in a balanced order such that all combinations occurred equally often. For three participants, data collected from one session was lost due to TMS-machine problems, so a third session was included. Participants were familiarised with the pictures and their correct names in each session. A short practice with 12 non-experimental trials with rTMS followed for familiarisation. The TMS-experiment consisted of 174 trials per session and lasted about 18 minutes. Presentation software (Neurobehavioral Systems, Inc., Albany, Calif., USA) controlled stimulus presentation and recording of responses. All participants wore ear plugs for protection from the TMS-induced noise.

Each picture appeared three times per list, once with each distractor type. Unique experimental lists (pseudorandomised and balanced) were generated from the two main lists for each participant. The same condition appeared at most three consecutive times and at least ten different pictures intervened between repetitions of the same picture.

Each trial began with a fixation cross for 2 sec (Fig 1B). The picture-word stimulus followed for 2 sec, with distractors displayed centred on the picture. rTMS was applied at image onset (see below). Participants were instructed to name the pictures as quickly and as correctly as possible, ignoring the distractors. Responses were recorded via a microphone connected to the Presentation PC and stored for analyses of response speed and accuracy.

**2.3.4 Transcranial magnetic stimulation.** We used frameless stereotaxy (Localite, Sankt Augustin, Germany) based on the coregistered individual T1-weighted magnetic resonance image (MRI) to navigate the TMS coil and maintain the exact location and orientation throughout the sessions. Individual structural T1-weighted scans were acquired in an extra session or taken from the institute's participant database (MPRAGE sequence in sagittal orientation, voxel size = 1 x 1 x 1.5 mm; TR = 1.3 s, TE = 3.36 ms; whole brain).

TMS was performed using the mean Montreal Neurological Institute (MNI) coordinates for left pSTG (-46, -30, 16) from a previous functional MRI study on picture-word-

interference with a similar design [20]. Vertex (Cz) was used as the control site because this area is a valid control condition, not associated with language processing [42–44]. The location of the vertex was determined manually as the midpoint between the lines connecting nasion and inion and tragi of the left and right ear in each subject as described previously [45].

Individual stimulation sites for pSTG were determined by calculating the inverse of the normalization transformation and transforming the coordinates from standard to individual space. During each session, subjects were co-registered to their individual structural scan.

Individual resting motor threshold (RMT) was defined as the lowest stimulation intensity producing a visible motor evoked potential of 50 µV (peak-to-peak amplitude) or greater in the relaxed first dorsal interosseus muscle in 5 out of 10 trials with single pulse TMS given over the motor hand area in the left primary motor cortex [46].

Stimulation intensity was corrected for the scalp-to-cortex distance between the motor cortex and the two stimulation sites using the following simple linear correction approach recommended by Stokes et al. [47]: $AdjMT\% = MT + 3 (D_{pSTG} - D_{M1})$.

where AdjMT% corresponds to the adjusted motor threshold in percentage stimulator output, MT is the unadjusted MT in percentage stimulator output, $D_{pSTG}$ is the distance between scalp and target in the left pSTG, and $D_{M1}$ corresponds to the distance between scalp and target in the motor cortex. The difference in the distance between the two sites is multiplied by 3 to account for the spatial gradient relating MT to distance [47]. Note that the distance correction was applied to 90% RMT (instead of 100% as in the original paper by Stokes et al., [47]) to avoid unpleasantly high stimulation intensities. A similar procedure has been applied in some of our previous TMS studies [25, 48, 49].

For the primary motor cortex, we used the mean stereotactic coordinates from a meta-analysis [50] as a starting point for the determination of the individual motor hotspot and applied the same algorithms as described above. For pSTG, the coil was oriented 45˚ to the sagittal plane, with the second phase of the biphasic pulse inducing a posterior-to-anterior current flow [24]. Due to anatomical restrictions, coil placement required rotation at an angle of 225˚ in some subjects. Consequently, the current flow was inversed in these subjects, assuring a similar current flow as in the original 45˚ condition. A similar procedure was used in our previous study that targeted temporal areas [48]. Vertex was defined as Cz by the 10–20 EEG system as described above. For vertex TMS, the handle was pointing backwards [51]. The position of the TMS coil was monitored during the whole experiment and adjusted if necessary. TMS was applied using a Magpro X100 stimulator (MagVenture, Farum, Denmark) and a figure-of-eight-shaped coil (C-B60; outer diameter 7.5 cm).

Across the experiment, an online TMS burst of five pulses at 10 Hz was applied at each picture onset (Fig 1B and 1C) at every trial. We used short bursts of 5 pulses as in our previous work in the language system [e.g. 25, 45] to assure sufficient power of the stimulation effect. Such high-frequency online rTMS bursts typically affect cortical activity in the stimulated area for a period outlasting the stimulation for about half the duration of the stimulation train [52], and thus provide a temporal resolution in the range of hundreds of milliseconds [53, 54], which would correspond to about 750 ms in our study. The timing estimates of different word production stages are debated in the literature, but authors largely agree that lexical selection and phonological encoding happen within a couple of hundreds of milliseconds after conceptual preparation [13, 55]. Therefore, we are confident that online rTMS should allow us to probe whether left pSTG is crucial for lexical selection.

TMS intensity was set to 90% of individual RMT of the left primary motor hand area and then corrected for the individual scalp-to-cortex distance [24] as described above. This resulted in a mean stimulation intensity of 38% maximum stimulator output across subjects (SD = 6.51). RMT was determined in the first session and held constant across both sessions to

guarantee that the same intensity was used for both TMS sites and all conditions within each participant as in our previous studies [e.g. 25, 45, 56, 57]. Overall TMS application and stimulation intensities were well within the published safety guidelines [58, 59].

**2.3.5 Analysis.** Vocal responses were examined offline for dysfluencies or errors. All trials associated with these errors were coded as incorrect and excluded from the response time (RT) analyses, in addition to trials with RT > 3 s (1 trial in total). Trials with technical failures were discarded (at most 10 trials for each distractor by list by stimulation site combination). Discarded and incorrect trials were equally distributed across conditions (X-squared = 0.004, df = 6, p = 1). Naming RTs were calculated manually relative to picture onset (by LN) from the speech signal before trials were split by the factors of interest, i.e., the experimenter was blinded for the condition a trial pertained to. Data were analysed in R [R Development Core Team, 2014; tidyverse and ggplot packages, 60, 61]. Single-trial RT and accuracy were analysed with linear and logistic mixed-effects models, respectively. Models were fitted with the lme4-package [33]. The RT model had the following structure (similar to the pre-test): as fixed effects, we used distractor condition, list, stimulation, the pictures' name agreement and frequency, and the 3-way interaction between list, stimulation, and condition, and the interaction between name agreement and list [given the importance of name agreement for naming latencies, 34, and the findings of the pre-test]. By-participant and by-item random intercepts were used. Condition, list, and stimulation were entered as repeated contrasts [35; MASS package, 36]. The error model using the same structured failed to converge. We reduced the model's complexity until convergence was achieved, resulting in one model per list, which included fixed effects of condition and stimulation (and for list 2 only, also their interaction) and by-participant random intercepts. Significance of effects was obtained using the Satterthwaite approximation [lmerTest-package, 37]. Data and analysis scripts are available at https://tinyurl.com/y5vky8y3.

## 3 Results

Fig 2 shows the RT results per list and Table 4 presents the details of the statistics. Overall RTs were 890, 858, and 765 ms for the related, unrelated, and congruent conditions, respectively (for details, see S1 Table in S1 File). Semantic interference (i.e., related trials were slower than unrelated trials) and congruency facilitation (i.e., congruent trials were faster than unrelated trials) effects were observed ($p$s < .001). Similar to the pre-test, there was a name agreement effect for list 2 ($p$ = 0.007), such that pictures with lower name agreement were named more slowly. The congruency effect differed as a function of list and stimulation site ($p$ < .001). We examined this three-way interaction further by testing the interaction between stimulation site and distractor condition for each list separately. The rest of the model structures was kept the same as for the main analyses (Table 5). For list 1, only the distractor effects were significant (i.e., related and congruent conditions relative to the unrelated condition, $p$s < 0.001). For list 2, the distractor effects were also significant ($p$s < 0.001). Importantly, the congruency effect (i.e., congruent vs unrelated trials) differed as a function of stimulation site ($p$ = 0.001). pSTG stimulation (salmon line in Fig 2, right panel), relative to vertex, further decreased the RTs in the congruent condition relative to the unrelated condition (RTs for congruent with vertex stimulation: 814 ms; pSTG stimulation: 751 ms). In a separate model examining the congruent condition only, with the fixed effects of stimulation, list, and their interaction, name agreement (in interaction with list), and frequency confirmed the stimulation effect (B = -12.03, S.E. = 5.09, t = -2.36, p = 0.018).

The comparisons had the following effect sizes (Cohen's d, calculated for each comparison separately): semantic interference effect with vertex stimulation, list 1: d = 0.846 and list 2:

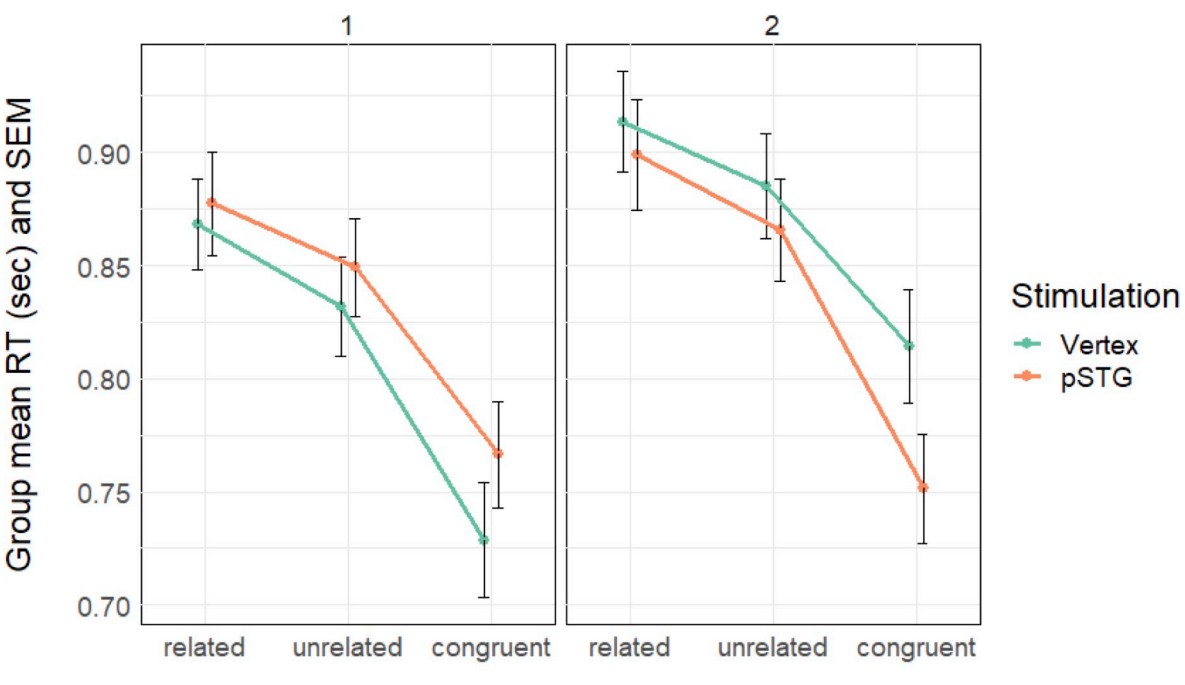

**Fig 2. Response time results.** Group-level response times (RT), calculated by computing the mean over participant-level means, and standard error of the mean (SEM) for vertex stimulation (green) and posterior superior temporal gyrus stimulation (pSTG, salmon) for list 1 (left) and list 2 (right).

**Table 4. Inferential statistics for the response times.**

| Condition / parameter | B | SE | t | p |
|---|---|---|---|---|
| Intercept | 894.34 | 29.57 | 30.25 | <0.001 |
| Unrelated vs related | -32.33 | 4.61 | -7.01 | <0.001 |
| Congruent vs unrelated | -94.12 | 4.57 | -20.59 | <0.001 |
| pSTG stimulation vs vertex | -5.67 | 3.75 | -1.51 | 0.131 |
| List 2 vs list 1 | 59.09 | 41.82 | 1.41 | 0.158 |
| Frequency | -0.19 | 0.11 | -1.68 | 0.092 |
| Unrelated vs related: pSTG vs vertex | 2.45 | 9.22 | 0.27 | 0.790 |
| Congruent vs unrelated: pSTG vs vertex | -10.48 | 9.14 | -1.15 | 0.252 |
| Unrelated vs related: list 2 vs list 1 | 0.72 | 9.22 | 0.08 | 0.937 |
| Congruent vs unrelated: list 2 vs list 1 | 1.10 | 9.14 | 0.12 | 0.904 |
| pSTG vs vertex: list 2 vs list 1 | -52.77 | 83.92 | -0.63 | 0.529 |
| List [1]: name agreement | -0.43 | 0.37 | -1.17 | 0.241 |
| List [2]: name agreement | -0.77 | 0.29 | -2.69 | 0.007 |
| Unrelated vs related: pSTG vs vertex: list 2 vs list 1 | -12.69 | 18.44 | -0.69 | 0.491 |
| Congruent vs unrelated: pSTG vs vertex: list 2 vs list 1 | -64.61 | 18.28 | -3.53 | <0.001 |

The linear regression model included by-participant and by-item random intercepts. B = beta coefficient;
SE = standard error; pSTG = mid-to-posterior superior temporal gyrus.

**Table 5. Inferential statistics for the response times for each list separately.**

| Condition/parameter | List 1 | | | | List 2 | | | |
|---|---|---|---|---|---|---|---|---|
| | B | SE | t | p | B | SE | t | p |
| Intercept | 863.55 | 37.59 | 22.97 | <0.001 | 923.64 | 35.98 | 25.67 | <0.001 |
| Unrelated vs related | -32.84 | 6.29 | -5.22 | <0.001 | -31.84 | 6.62 | -4.81 | <0.001 |
| Congruent vs unrelated | -94.30 | 6.24 | -15.11 | <0.001 | -93.73 | 6.56 | -14.28 | <0.001 |
| pSTG stimulation vs vertex | 20.52 | 43.96 | 0.47 | 0.641 | -32.01 | 46.19 | -0.69 | 0.488 |
| Name agreement | -0.43 | 0.33 | -1.30 | 0.193 | -0.76 | 0.32 | -2.36 | 0.018 |
| Frequency | -0.14 | 0.18 | -0.79 | 0.430 | -0.21 | 0.15 | -1.39 | 0.166 |
| Unrelated vs related: pSTG vs vertex | 8.96 | 12.58 | 0.71 | 0.477 | -3.77 | 13.25 | -0.28 | 0.776 |
| Congruent vs unrelated: pSTG vs vertex | 21.93 | 12.48 | 1.76 | 0.079 | -42.82 | 13.13 | -3.26 | 0.001 |

The linear regression model included by-participant and by-item random intercepts. B = beta coefficient; SE = standard error; pSTG = mid-to-posterior superior temporal gyrus; NA = not available.

d = 2.098; with pSTG stimulation, list 1: d = 1.127 and list 2: d = 0.908; congruency facilitation effect, with vertex stimulation, list 1: d = 2.170 and list 2: d = 1.923; with pSTG stimulation, list 1: d = 3.514 and list 2: d = 3.187.

The overall error rate was 2.31%. Mean error rates per list, distractor condition, and stimulation site are presented in S2 Table in S1 File. S3 Table in S1 File presents the details of the statistics for the errors. In summary, for list 1, a congruency effect was found ($p < 0.001$). For list 2, both semantic and congruency effects were significant ($ps < 0.006$).

We tested post-hoc for the presence of a speed-accuracy trade-off pattern in the congruent condition in list 2 as a function of stimulation site in two ways. Firstly, we correlated participants' mean RTs with their mean accuracy for each condition (list by stimulation site by distractor type) separately. The scatterplots of the relation between mean RT and mean accuracy are shown in S1 Fig in S1 File. None of the correlations, assessed with a non-parametric Spearman correlation test, were significant (all $ps > 0.220$), indicating that the participants who were faster were not the participants committing more errors. Moreover, using a linear regression, we tested whether mean RT could be explained by mean accuracy in interaction with stimulation site for the congruent condition in list 2. Under the assumption that there is a difference in speed-accuracy trade-off as a function of stimulation site, an interaction should be obtained. However, mean accuracy, stimulation site, or their interaction were not significant predictors in the model (all $ps > .275$). Hence, a speed-accuracy trade-off does not provide a complete account for our findings.

## 4 Discussion

Here, we investigated the causal role of the left pSTG in processing interference from distractor words in word production, using a picture-word interference paradigm with congruent, categorically related, and unrelated distractors. Based on previous lesion and neuroimaging studies [14, 16, 18, 20, 62], we reasoned that focal perturbation of this area should interfere with processing stages of planning the target word, thereby increasing the semantic interference effect. Surprisingly, a different pattern emerged. We found that rTMS selectively *facilitated* responses in the *congruent* condition in the somewhat more difficult list 2. Despite the lists being matched on important variables, name agreement in list 2 emerged as an important predictor of naming latencies, likely affecting word production stages at the semantic, lexical, and phonological levels. In the congruent condition, this may have interacted with the distractor word, rendering it less converging at all these levels with the picture (i.e., the "congruent" distractor

word may not be the participant's preferred label for a given picture and, in this regard, the distractor is no longer congruent at the lexical and phonological levels). The selective beneficial rTMS effect in the congruent condition in list 2 suggests that stimulation of the left pSTG facilitated the (convergence of) processing streams for picture naming and word reading under increased task difficulty only, likely because the response speed for the congruent condition was already at ceiling for the easier list 1. This pattern could not be explained by a speed-accuracy trade-off. This speaks for an impact of the task- or stimulus-induced brain state, as we discuss below.

Our left pSTG stimulation site was based on a previous study that found increased cross-domain activity at this site for categorically related stimuli in cognitive control tasks [20; similar coordinates obtained by 14, 16]. We reasoned that perturbation of this area should induce noise in the stimulated area that would affect the activation of representations of both target and distractors [18]. Note that although the exact physiological mechanisms underlying neuro-stimulation effects are unclear, it is argued that the excitation of random neural elements by online TMS likely causes neuronal noise in the stimulated area [63], which may impair or delay task-relevant neuronal computations because neural activity needs to be sampled longer to discriminate signal and noise [64]. Accordingly, we reasoned that this should increase difficulty in resolving conflict between picture and distractor representations, resulting in stronger semantic interference for the related condition, contrary to what we found.

We note that the functional MRI studies that we used to motivate our target selection [14, 16, 20] lack temporal resolution. Thus, the evidence provided by those studies pertains to the brain areas modulated by properties of the distractors *overall*, while the extent to which they can inform us on particular stages of production remains somewhat limited. Given the relatively long TMS burst that was applied to interfere with task processing, the present study also lacks the temporal resolution to link the pSTG to one particular processing stage. However, the observed task-specificity of the TMS effect in the congruent condition, in particular for a stimulus list with more difficult items (possibly due to name agreement), does suggest that left pSTG played a role in participants' performance during picture-word interference, a task in which the representations of picture and distractor (or their processing streams) interact. Future studies are needed to link the role of left pSTG to a particular stage of word production.

When considering the observed facilitation effect, it should be noted that the direction of 10 Hz rTMS protocols in the study of cognition is far from clear. We initially expected an rTMS-induced *delay* in RTs because the majority of previous studies in the language domain reported prolonged RTs during different language comprehension tasks when rTMS was applied shortly after visual or auditory word onset [e.g. 24, 26, 27, 65]. However, high-frequency rTMS does not necessarily lead to behavioral inhibition [64]. Indeed, other language production studies found facilitation when the same rTMS protocol was applied over the inferior frontal or posterior temporal cortex immediately before stimulus presentation [66–69] or shortly after stimulus onset [45]. Most of these studies used simple picture naming paradigms that would be most comparable with our congruent condition. Notably, behavioural facilitation in language comprehension tasks was further observed when 10 Hz rTMS was applied over the left pSTG in an offline fashion, that is, prior to task processing [e.g. 70, 71], which argues for a strong impact of the timing of the stimulation protocol.

With respect to the underlying mechanisms of facilitatory rTMS effects, Miniussi, Ruzzoli and Walsh [72] argued that the rTMS-induced activity in the stimulated area can be considered both as noise and as part of the task-related signal, depending on the activated neuronal population. The induced activity might be synchronized with the ongoing relevant signal, thereby rendering the signal stronger and providing an "optimal" level of noise for a specific task or process, which might explain the observed facilitation effect in several studies of

cognition [73], including the present one. Moreover, factors like stimulation intensity, time point of stimulation, and task difficulty have been shown to affect behavioral outcome, particularly in online rTMS paradigms like the one we used [74, 75]. Consequently, the impact of an rTMS-induced perturbation might change with varying task conditions and complexity [76]. Accordingly, when rTMS is applied to a region expected to be involved in a cognitive process immediately before the process is executed, the initial neuronal activation state of that region is altered, causing divergent behavioral effects [77, 78]. In our study, 10 Hz rTMS might have increased the amount of activity in the targeted area to a level that was optimal for task performance, potentially resulting in a "pre-activation" of task-relevant activity [for a similar reasoning, see 45, 67, 68]. Indeed, a meta-analysis has associated activity in the left pSTG in a time window between 275–400 ms after stimulus onset with phonological-code retrieval during picture naming [13]. Given that our stimulation protocol started with stimulus onset, it is reasonable to assume that pre-activation might have already occurred prior to phonological encoding of the response.

As the congruent condition does not require the participants to ignore distractors, one may argue that the observed facilitation might be explained by unspecific effects, such as inter-sensory facilitation induced by our stimulation protocol [e.g., 79]. However, this explanation would be incompatible with our interpretation of specific effects that facilitated task-related processing in the targeted area. We argue that the high specificity of our facilitation effect, selectively observed with increasing task demands in the congruent condition, argues against the unspecificity hypothesis. Rather, we believe that our findings indicate a specific facilitation effect best explained within the state-dependency framework [74, 75]. Within this framework, it is argued that the direction and strength of an rTMS-induced effect strongly depends on the task-induced brain state and might thus either result in inhibition or facilitation of task performance. While our two stimulus lists were matched with respect to lexicality, phonological familiarity and regularity and word length, they still differed in terms of task difficulty, most likely at the level of picture name agreement as determined post-hoc by our statistical analyses. In our design, the distractor word should prime the picture name and vice versa. We speculate that low name agreement in the congruent condition may translate into less congruency between picture name and distractor and thus less priming. Within the framework of state-dependency, it is possible that TMS might have facilitated the less active neural populations during the task [see 80, 81], which may explain the observed selective facilitation of the congruent condition in the somewhat more difficult list (based on a significant effect of name agreement only in list 2). However, it remains unclear why TMS selectively facilitated the congruent condition in list 2 without modulating the more difficult unrelated or related distractor conditions. A likely explanation for this finding is that name agreement had the relatively strongest impact on the congruent condition where picture and distractor usually converge, but less so when name agreement is low. Accordingly, we observed a significant effect of pSTG TMS in the congruent condition for list 2 only, which may have resulted from a facilitation of the matching between word and picture under conditions with "increased cognitive load" (as reflected in overall longer naming latencies due to the name agreement effect in list 2).

This study has a number of limitations. Our choice for the statistical method with a complex structure resulted in convergence problems for the full model of the error data. Therefore, list 1 and 2 could not be compared directly. However, since the most relevant and informative findings regard the RTs, for which these comparisons could be made, the lack of comparison for the errors is not a major limitation. Another limitation is the fact that, due to design constraints, only half of the participants received rTMS over the pSTG with list 2. Thus, our findings for the congruency effect for list 1 could be a cohort effect. Future studies replicating the present findings are needed. Due to technical issues, three participants had to undergo a third

TMS session. In one participant, vertex TMS with list 1 had to be repeated, in two other participants, pSTG TMS was repeated with list 1. However, given the small number of repetitions per TMS conditions, we are confident that this did not systematically influence our results. Note that the list effect (i.e., list 1 easier than list 2) was already present in the pre-test. Finally, with respect to the TMS procedure, it should be noted that TMS may induce blinking in participants which may interfere with viewing stimuli or other side effects such as face twitches. Moreover, the acoustic stimulation has been demonstrated to induce cross-modal resetting of occipital alpha oscillations [e.g., 82]. We are confident that these issues are unlikely to have strongly impacted our results. Even if the clicking sound might have influenced cross-modal resetting of oscillations or muscle twitching was present, this should have a similar impact on all task conditions and would thus be unlikely to explain our findings.

In light of our unexpected findings, we wish to emphasize that the specific contribution of different temporal-lobe regions to specific stages of word production is still unclear. Future studies might explore these issues by applying single pulse TMS or short bursts at different time points during the task. Such approaches have provided insight into the time-course of the involvement of frontal and temporal areas during picture naming [e.g., 83]. We wish to emphasize that we did not have a-priori hypotheses regarding the time-specific role of the pSTG during picture-word interference and chose a long stimulation period to assure effective modulation of task processing. It would also be interesting to test whether the observed TMS-induced modulation may be different when targeting the (mid-to-posterior) middle temporal gyrus or the left inferior frontal gyrus, which represent further key nodes for language production [13]. Finally, future studies may zoom into the effects of different competitors and disentangle the time course of the involvement of different processes (e.g., lexical and phonological) by including both semantically and phonologically related distractors.

In conclusion, our study adds new evidence to the causal role of the left pSTG in the interaction effect between picture and distractor representations or processing streams, only partly supporting previous neuroimaging studies. Our findings support the idea that the task- or stimulus-induced brain state strongly interacts with the stimulation effect and that the timing of the stimulation protocol may be crucial with respect to the observed outcome (i.e., facilitation or inhibition). Consequently, these issues should be systematically addressed in future rTMS studies on language production.

## Supporting information

**S1 File.**
(DOCX)

## Author Contributions

**Conceptualization:** Vitória Piai, Gesa Hartwigsen.

**Formal analysis:** Vitória Piai.

**Funding acquisition:** Vitória Piai, Gesa Hartwigsen.

**Investigation:** Laura Nieberlein.

**Methodology:** Gesa Hartwigsen.

**Supervision:** Vitória Piai, Gesa Hartwigsen.

**Validation:** Vitória Piai.

**Visualization:** Vitória Piai, Gesa Hartwigsen.

**Writing – original draft:** Vitória Piai, Gesa Hartwigsen.

**Writing – review & editing:** Vitória Piai, Laura Nieberlein, Gesa Hartwigsen.

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
