## [Decision Letter · Decision Letter 0]

6 Oct 2020

PONE-D-20-24893

Effects of transcranial magnetic stimulation over the left posterior superior temporal gyrus on picture-word interference

PLOS ONE

Dear Dr. Hartwigsen,

Thank you for submitting your manuscript to PLOS ONE. After careful consideration, we feel that it has merit but does not fully meet PLOS ONE’s publication criteria as it currently stands. Therefore, we invite you to submit a revised version of the manuscript that addresses the points raised during the review process.

Both Reviewers find your study interesting and sound, however they have some concerns about some relevant missing information in the Manuscript and the possible blinking caused by the rTMS procedure. Please consider the comments of both Reviewers carefully. 

We look forward to receiving your revised manuscript.

Kind regards,

Nicola Molinaro, Ph.D.

Academic Editor

PLOS ONE

2. Please change "female” or "male" to "woman” or "man" as appropriate, when used as a noun.

Reviewers' comments:

Reviewer's Responses to Questions

**Comments to the Author**

1. Is the manuscript technically sound, and do the data support the conclusions?

Reviewer #1: Yes

Reviewer #2: Yes

2. Has the statistical analysis been performed appropriately and rigorously? 

Reviewer #1: Yes

Reviewer #2: Yes

3. Have the authors made all data underlying the findings in their manuscript fully available?

Reviewer #1: Yes

Reviewer #2: Yes

4. Is the manuscript presented in an intelligible fashion and written in standard English?

Reviewer #1: Yes

Reviewer #2: Yes

5. Review Comments to the Author

Reviewer #1: This study aimed to investigate the role of left pSTG in semantic interference. High-frequency online rTMS over the left pSTG was applied when participants performed a picture-word interference task. Based on results from previous neuroimaging studies, the authors predicted pSTG stimulation should interfere with word production stages, resulting in stronger semantic interference compared to stimulation over the vertex (control site). Contrary to this prediction, the effect of pSTG stimulation on semantic interference was not significant. Surprisingly, pSTG stimulation selectively increased the congruency effect in the more difficult list, that is, the facilitatory effect of congruent word-picture stimuli was further facilitated by the rTMS over the pSTG. This is an original and interesting study. It has been well conducted and the manuscript is well written.

Below I have raised a number of points that I would like to see addressed in a revised version of this manuscript to further improve it.

Introduction

I think the use of the term "virtual lesion" to describe the impact induced by high-frequency online rTMS is somewhat misleading and would suggest revision. As the authors note elsewhere, this protocol is likely to interfere temporarily with on-going processes by introducing noise at a critical period of time. The term “virtual lesion” is somewhat controversial but might be more applicable in any case in situations where the effects of rTMS are slightly longer lasting or in a treatment trial.

Pre-test

p.5 second paragraph: please explain how the H-index is calculated?

Please state that the words were presented in German.

p.8 “As expected, in the RTs, a congruency effect was found (unrelated vs. congruent, p < 0.001), as well as semantic interference (related vs. congruent, p < 0.001)”. It was my understanding that semantic interference refers to related vs. unrelated (see abstract for example). Please clarify.

Please clarify what is meant by “distractor effects” in each instance it is used. Is it always used to mean both Unrelated vs related AND Congruent vs unrelated?

TMS experiment

A major concern I have for the TMS experiment is the sensation difference between the stimulation and control areas. Compared to the control stimulation, pSTG stimulation is much more likely to stimulate the peripheral nerves (facial) as well, which might cause face twitch, especially around eyes and mouths. I was wondering whether the participants noticed the sensation differences. Were participants aware of the nature of the experiment. This should be discussed, if so.

It appears that rTMS was applied on every trial. Please confirm. A design with trials where there was no TMS would have been informative – please discuss.

The clarity of the results section could be improved as it was not always clear which comparison was being referred to. Also, although the results of linear mixed effects models were presented in tables of the results, these are not straightforward to interpret. The reader would benefit from some further interpretation in the text.

For example, from the results presented, it is unclear whether there is a difference between congruent and related for the pSTG vs vertex. I think the only way to present congruent vs related contrasts would be to rerun the model with a different reference category and present those results. Could the authors clarify?

The manuscript would benefit from a limitations section. For example, the use of linear mixed effects models with complex error structure resulted in eventual simplification of the model so that the two word lists were run separately. The effect of rTMS over pSTG to further increase the effect of congruency on RT was only evident for list 2 but we don’t know if this was a significant difference compared with list 1 as the two were not compared. Please discuss this and some other limitations of the study. Although the results were evident for list 2 and explained because list 2 was more difficult, the design would mean that only half the participants received rTMS over the pSTG during list 2 – so it could be a cohort effect. Three participants had a third session – which list was repeated and for which stimulation site?

The results are intriguing but would require further replication with some potential improvements to the design and perhaps simpler modelling.

Reviewer #2: In this study, the authors investigated the potential involvement of the left posterior STG (pSTG) in semantic interference. To this aim, they applied focal repetitive transcranial magnetic stimulation (rTMS at 10 Hz) to the aforementioned region while participants performed a picture-word interference task in which they had to name pictures presented with congruent, categorically related, or unrelated distractor words. Overall, they hypothesized that online stimulation over the left pSTG, should interfere with processes related to word production, resulting in an increased semantic interference effect (i.e., related vs. unrelated conditions) indexed by slower reaction times for pSTG stimulation relative to active control site (i.e., vertex). Overall, contrary to their initial hypothesis, pSTG stimulation did not affect the semantic interference effect but increased the congruency effect (i.e., faster responses for the congruent condition) in the more difficult stimuli (i.e., list of items with lower name agreement). This is an interesting study with a solid methodological and statistical approach. I only have one major concern and some minor points about the Methods section. See below.

General comment:

The authors show that pSTG stimulation leads to faster responses for congruent distractors in List 2 and interpret this effect as resulted from “a facilitation of the matching between word and picture under conditions with “increased cognitive load” and later on: “In our study, 10 Hz rTMS might have increased the amount of activity in the targeted area to a level that was optimal for task performance, potentially resulting in a “pre-activation” of task-relevant activity”. However, while observing error rates in Table S2 it is also true that, for the congruent condition in List 2, participants committed more errors after pSTG stimulation as compared to List 2-vertex stimulation (and also as compared to all congruent distractors irrespectively of list and site). Hence, this facilitatory pattern reflected in faster responses does not necessarily reflect a boosting in task performance (after stimulation participants might be faster but ALSO less accurate). I think this aspect is not properly stressed or discussed by the authors.

Methods:

1. How was the sample size estimated? Did the authors conducted any power analysis before running the experiment?

2. In the pretest (and TMS study) the authors used logistic and linear mixed effects models for modelling accuracy and RTs, respectively. Which is the rationale behind this choice?

3. Why participants were tested in different days? Furthermore, why do the authors use stimulation parameters acquired in day 1 for stimulation in day 2? The state of the stimulated region (baseline cortical activation) might have varied from one week to the other. Waiting for > 60min between both stimulations would have been enough to rule out that transient changes provoked by stimulation on one site interfered with stimulation of the second site.

4. For the stimulation protocol, was the site (pSTG, vertex) order counterbalanced across participants?

5. Vocal responses: how many trials were rejected (>3ms) or discarded due to technical issues? Did the number of trials significantly differ among conditions? Please report.

6. TMS may induce blinking in participants thus interfering with viewing stimuli. Furthermore, acoustic stimulation (“clicking” sound of the coil) is known to provide a significant cross-modal resetting of occipital alpha oscillations (e.g., Romei and colleagues work). Since the authors were using visual stimuli, to what extend do they think these aspects could have influenced the observed patter of results?

Discussion and interpretation of findings:

The authors mention that their findings are comparable/compatible with those observed in a study in the visual domain (Schwarzkopf, Silvanto & Rees, 2011), showing increased facilitation for more difficult targets. However, I don’t think this holds for the present study. It is true, that List 2 was somehow more difficult than List 1 but, in the overall context of the experiment, the congruent condition is still the easiest one, as shown by faster overall responses as compared to other conditions irrespectively of the TMS session.

The authors argue that their findings can be interpreted in light of the state-dependent approach and I completely agree with them. Here, the authors used a task in which the distractor is supposed to prime the picture name. If low name agreement in the congruent condition translates into less congruency between picture name and distractor (and thus less priming), then it is possible that TMS might have facilitated the less active neural populations during the task (e.g., see Cattaneo et al. 2008 and Silvanto & Pascual-Leone, 2008 for a review).

---

## [Author Response · Author response to Decision Letter 0]

29 Oct 2020

PONE-D-20-24893-R1

Effects of transcranial magnetic stimulation over the left posterior superior temporal gyrus on picture-word interference

Response To Reviewers

Reviewer #1: 

Comment 1. ”This is an original and interesting study. It has been well conducted and the manuscript is well written. Below I have raised a number of points that I would like to see addressed in a revised version of this manuscript to further improve it.”

Reply 1. We would like to thank the reviewer for the positive evaluation of our manuscript. We hope that we have sufficiently addressed all the reviewer’s concerns as detailed below.

Comment 2. “I think the use of the term "virtual lesion" to describe the impact induced by high-frequency online rTMS is somewhat misleading and would suggest revision. As the authors note elsewhere, this protocol is likely to interfere temporarily with on-going processes by introducing noise at a critical period of time. The term “virtual lesion” is somewhat controversial but might be more applicable in any case in situations where the effects of rTMS are slightly longer lasting or in a treatment trial.”

Reply 2. We agree with the reviewer that the term “virtual lesion” is unfortunate and might cause confusion, especially when used for online TMS studies like the present one. Consequently, we have consistently replaced “virtual lesion” with “perturbation” or “TMS-induced perturbation” throughout the manuscript.

Comment 3. “p.5 second paragraph: please explain how the H-index is calculated?” 

Reply 3. The following information has been added, please see pages 5-6.

“The H-index for each picture was calculated as

H= ∑_(i=1)^k▒〖p_i 〖log〗_2 (1/p_i ) 〗

where k refers to the number of different names given to a picture and pi to the proportion of participants that gave each one of the names.”

Comment 4. “Please state that the words were presented in German.”

Reply 4. This information has been added to page 5 and page 11:

“The picture names were high-frequency nouns in German. Pictures were coloured photographs on a white background (28; and from the internet). German distractors were derived from the picture names […]”

“For the TMS study, the controlled German materials of the pretest were widely adopted.”

Comment 5. “p.8 “As expected, in the RTs, a congruency effect was found (unrelated vs. congruent, p < 0.001), as well as semantic interference (related vs. congruent, p < 0.001)”. It was my understanding that semantic interference refers to related vs. unrelated (see abstract for example). Please clarify.” 

Reply 5. The reviewer is correct, we apologise for the mistake. The revised sentence on page 8 reads as follows: “As expected, in the RTs, a congruency effect was found (i.e., responses were slower for unrelated than for congruent trials, p < 0.001), as well as semantic interference (i.e., responses were slower for related than for unrelated trials, p < 0.001).” 

and 

“In the errors, a congruency effect was found (i.e., more errors for unrelated than for congruent trials, p < 0.001), as well as semantic interference (i.e., more errors for related than for unrelated trials, p = 0.033).”

Comment 6. “Please clarify what is meant by “distractor effects” in each instance it is used. Is it always used to mean both Unrelated vs related AND Congruent vs unrelated?” 

Reply 6. This has been clarified on page 9: “[…] the expected distractor effects (i.e., congruent and related conditions differ from the unrelated condition)”.

and on page 17: “For list 1, only the distractor effects were significant (i.e., related and congruent conditions relative to the unrelated condition, ps < 0.001).” 

TMS experiment 

Comment 7. “A major concern I have for the TMS experiment is the sensation difference between the stimulation and control areas. Compared to the control stimulation, pSTG stimulation is much more likely to stimulate the peripheral nerves (facial) as well, which might cause face twitch, especially around eyes and mouths. I was wondering whether the participants noticed the sensation differences. Were participants aware of the nature of the experiment. This should be discussed, if so.” 

Reply 7. The reviewer raises an important issue. In this study, we did not include a formal unpleasantness rating as used in some of our earlier rTMS studies, which used similar 10 Hz online protocols over prefrontal and parietal regions (Hartwigsen et al., 2010a,b) because we noted that this may prime participants towards expecting something unpleasant. In the present study, none of the subjects reported side effects or correctly identified the effective TMS session but we did not include a formal questionnaire on this either. Most importantly, we wish to emphasize that even if some subjects had noticed sensation differences between vertex and pSTG TMS, this difference was kept constant across the three task conditions. Moreover, if pSTG TMS was more unpleasant, one would expect a stronger disruption of task performance, rather than the observed facilitation due to these side effects. This point is now discussed in the limitations paragraph on page 25:

“Finally, with respect to the TMS procedure, it should be noted that TMS may induce blinking in participants which may interfere with viewing stimuli or other side effects such as face twitches. [...] Even if the clicking sound might have influenced cross-modal resetting of oscillations or muscle twitching was present, this should have a similar impact on all task conditions and would thus be unlikely to explain our findings.” 

Comment 8. “It appears that rTMS was applied on every trial. Please confirm. A design with trials where there was no TMS would have been informative – please discuss.” 

Reply 8. Indeed, rTMS was applied on every trial, this information has been made explicit on page 15: “Across the experiment, an online TMS burst of five pulses at 10 Hz was applied at each picture onset (Figure 1B, C) at every trial.” 

With respect to the inclusion of no-TMS trials, we are aware that previous studies included such a condition, sometimes interleaved with effective TMS to control for carry-over or practice effects (e.g. Sandrini et al., 2011 for review). We decided against this for several reasons. First, a no-TMS condition does not control for any unspecific side effects of the stimulation procedure. Secondly, the difference between TMS and no-TMS will be obvious to the participants, which may confound the results. Consequently, it was argued that the inclusion of an active control site is always preferable and that the duration of intertrial intervals and/or TMS trains may be adopted to avoid carry-over effects between trials or conditions (Bergmann and Hartwigsen, 2020). We are confident that carry-over effects between trials are not an issue in our study since the intertrial-interval was long enough to prevent them. High-frequency online rTMS bursts like the one applied in our study are thought to affect cortical activity in the stimulated area for a period outlasting the stimulation for about half the duration of the stimulation train (Rotenberg et al., 2014). Given that we applied 5 pulses with picture onset, the after-effect should be too short-lasting to interfere with the upcoming trial, because each picture-word pair was presented for 2 s and the next trial started with a fixation cross, which was presented for another 2 s. In summary, we do not think that the inclusion of a no-TMS condition would have been informative in our study.

Comment 9. “The clarity of the results section could be improved as it was not always clear which comparison was being referred to. Also, although the results of linear mixed effects models were presented in tables of the results, these are not straightforward to interpret. The reader would benefit from some further interpretation in the text.” 

Reply 9. We have now made more explicit which conditions are being compared. Moreover, we have attempted to describe the results more clearly. However, we do not believe that making explicit how each model parameter should be interpreted would improve clarity. For example, when a regular ANOVA is performed, the F value is not explicitly interpreted either. However, if the reviewer would like to indicate what should be written more explicitly, we are happy to implement it.

Comment 10. “For example, from the results presented, it is unclear whether there is a difference between congruent and related for the pSTG vs vertex. I think the only way to present congruent vs related contrasts would be to rerun the model with a different reference category and present those results. Could the authors clarify?” 

Reply 10. The reviewer is correct that the only way to present the congruent vs related contrast would be to rerun all models. However, the related vs congruent contrast is of smaller theoretical relevance for the present study than the contrasts we report, and for the congruency effect as “unrelated vs congruent” in particular. This is because the related distractor is incongruent with the picture while strongly sharing a semantic relationship, whereas the unrelated distractor is also incongruent with the picture, while being a neutral lexical distractor. Therefore, the contrasts with the unrelated condition are to be preferred since they maximise the semantic (vs related) and congruency (vs congruent) effects. Since we make the data and analysis code available, the interested reader can inspect those contrasts as well. However, to keep the conciseness of our study, we focus on the two theoretically relevant contrasts. 

Comment 11. “The manuscript would benefit from a limitations section. For example, the use of linear mixed effects models with complex error structure resulted in eventual simplification of the model so that the two word lists were run separately. The effect of rTMS over pSTG to further increase the effect of congruency on RT was only evident for list 2 but we don’t know if this was a significant difference compared with list 1 as the two were not compared. Please discuss this and some other limitations of the study. Although the results were evident for list 2 and explained because list 2 was more difficult, the design would mean that only half the participants received rTMS over the pSTG during list 2 – so it could be a cohort effect. Three participants had a third session – which list was repeated and for which stimulation site?” 

Reply 11. We thank the reviewer for this suggestion, a limitations section is certainly valuable. We now discuss the suggested points in the limitations section on pp. 24-25. 

We would like to clarify that the issue of not being able to compare the two lists directly only holds for the errors, not for the RTs. For the RTs, as we originally reported on page 17, the effect of pSTG stimulation (vs vertex) is confirmed for the congruent condition across the two lists: “In a separate model examining the congruent condition only, with the fixed effects of stimulation, list, and their interaction, name agreement (in interaction with list), and frequency confirmed the stimulation effect (B = -12.03, S.E. = 5.09, t = -2.36, p = 0.018).” When we examine the three-way interaction between list, stimulation site, and the two conditions of the congruency effect (congruent vs unrelated) for the RTs in a separate model, the three-way interaction is again significant (p < 0.001). To further interpret this three-way interaction, one factor needs to be taken out of the model (by inspecting separate levels), which we do by either looking at the congruent condition only (originally reported, which confirms the stimulation effect) or by looking at the congruency * stimulation effect within each list, which confirms the stimulation effect on the congruency effect only for list 2 (p = 0.001). Therefore, in the limitations section, we only discuss the limitation of model complexity with respect to the errors. 

With respect to the three participants who had to undergo a third session, we have added the following information to the limitations section (page 25): “Due to technical issues, three participants had to undergo a third TMS session. In one participant, vertex TMS with list 1 had to be repeated, in two other participants, pSTG TMS was repeated with list 1. However, given the small number of repetitions per TMS conditions, we are confident that this did not systematically influence our results. Note that the list effect (i.e., list 1 easier than list 2) was already present in the pre-test.” Even though these three repetitions always involved list 1, which would explain the facilitation effect of list 1 relative to list 2, we note that the list effect (with list 1 being less difficult) was already present in the pre-test. Therefore, this repetition of list 1 in the TMS experiment cannot (fully) account for our findings. 

References

Bergmann TO, Hartwigsen G. (2020). Inferring Causality from Noninvasive Brain Stimulation in Cognitive Neuroscience. J Cogn Neurosci,1-29. doi: 10.1162/jocn_a_01591. Online ahead of print.

Hartwigsen G, Baumgaertner A, Price CJ, Koehnke M, Ulmer S, Siebner HR. (2010a). Phonological decisions require both the left and right supramarginal gyri. Proc. Natl. Acad. Sci. USA 107, 16494–16499. 

Hartwigsen G, Price CJ, Baumgaertner A, Geiss G, Koehnke M, Ulmer S, Siebner, HR. (2010b). The right posterior inferior frontal gyrus contributes to phonological word decisions in the healthy brain: evidence from dual-site TMS. Neuropsychologia 48, 3155–3163. 

Rotenberg, A., Horvath, J.C., Pascual-Leone, A. (Eds.), 2014. Neuromethods: Transcranial Magnetic Stimulation. Springer, New York, NY. http://dx.doi.org/10.1007/978-1-4939-0879-0.

Sandrini M, Umilta C, Rusconi E. (2011). The use of transcranial magnetic stimulation in cognitive neuroscience: a new synthesis of methodological issues. Neurosci Biobehav Rev 35, 516-536.

Reviewer #2: 

Comment 1: “This is an interesting study with a solid methodological and statistical approach. I only have one major concern and some minor points about the Methods section. The authors show that pSTG stimulation leads to faster responses for congruent distractors in List 2 and interpret this effect as resulted from “a facilitation of the matching between word and picture under conditions with “increased cognitive load” and later on: “In our study, 10 Hz rTMS might have increased the amount of activity in the targeted area to a level that was optimal for task performance, potentially resulting in a “pre-activation” of task-relevant activity”. However, while observing error rates in Table S2 it is also true that, for the congruent condition in List 2, participants committed more errors after pSTG stimulation as compared to List 2-vertex stimulation (and also as compared to all congruent distractors irrespectively of list and site). Hence, this facilitatory pattern reflected in faster responses does not necessarily reflect a boosting in task performance (after stimulation participants might be faster but ALSO less accurate). I think this aspect is not properly stressed or discussed by the authors.”

Reply 1: We thank the reviewer for the positive evaluation of our manuscript and the very constructive and helpful feedback. It is true that, for the congruent condition, list 2 with pSTG stimulation descriptively resulted in the largest number of errors overall. 

We have directly tested whether participants were faster at the cost of being less accurate. For that, we correlated participants’ mean response times with their mean accuracy for each condition (distractor by stimulation by list). We have added these results to the Results section and Supplementary Figure S1, which shows the scatterplots for the relationship between accuracy and response times. None of the correlations were significant (all ps > 0.220). Moreover, using a linear regression, we tested whether mean RT could be explained by mean accuracy in interaction with stimulation site for the congruent condition in list 2. Under the assumption that there is a difference in speed-accuracy trade-off as a function of stimulation site, an interaction should be obtained. However, mean accuracy, stimulation site, or their interaction were not significant predictors in the model (all ps > .275). Hence, a speed-accuracy trade-off does not provide a complete account for our findings. 

Methods: 

Comment 2. “How was the sample size estimated? Did the authors conducted any power analysis before running the experiment?” 

Reply 2. We did not perform an a-priori power analysis to determine the desired sample size. Our sample size was determined on the basis of feasibility, on the one hand, and on the basis of the sample sizes of the previous articles we based our study on (i.e., Abel et al., 2012; de Zubicaray et al., 2013; Piai et al., 2013, with 19, 20 and 23 participants respectively). Please note that the inclusion of 24 subjects allowed us to counterbalance all relevant factors (i.e., order of TMS, list and conditions). Please note that our recent TMS studies included similar sample sizes (n=24; e.g. Klaus & Hartwigsen, 2019; Kuhnke et al., 2017; Kroczek et al., 2019). 

Comment 3. “In the pretest (and TMS study) the authors used logistic and linear mixed effects models for modelling accuracy and RTs, respectively. Which is the rationale behind this choice?” 

Reply 3. The rationale is that, with continuous data (RTs), a linear regression can be applied, but with accuracy (correct or incorrect, a binary outcome), a linear regression should not be used since the binary outcome violates the assumption of a continuous outcome. This information has been added to the analysis section on page 7. 

Comment 4. “Why participants were tested in different days? Furthermore, why do the authors use stimulation parameters acquired in day 1 for stimulation in day 2? The state of the stimulated region (baseline cortical activation) might have varied from one week to the other. Waiting for > 60min between both stimulations would have been enough to rule out that transient changes provoked by stimulation on one site interfered with stimulation of the second site.” 

Reply 4. The reviewer raises an important issue. The brain state might indeed vary from one day to another, but this variation should affect all task conditions. We decided to implement different sessions with a constant stimulation intensity for the following reasons: First, performing the whole experiment with both TMS sessions on the same day would have resulted in overly long sessions. Please note that the first session may take up to 1.5-2 hours in some participants because threshold determination and coil placement with the neuronavigation system can be tricky. Consequently, in our TMS studies, we experienced that participants get tired if the session duration is too long, even if breaks are included. Therefore, we prefer to invite them to different sessions. Secondly, we prefer a constant stimulation intensity across conditions in our TMS studies to avoid systematic differences in the intensity between conditions of interest (i.e., area of interest and control site). The TMS intensity is calibrated to the individual excitability of the primary motor hand area by using the motor threshold as reference, mainly because of a lack of more appropriate procedures (see Hartwigsen, 2015). Yet, it remains unclear for which cortical areas the motor threshold of the primary motor cortex works as a good predictor in terms of regional excitability (Boroojerdi et al., 2002). Since the motor threshold is not necessarily a good predictor of the excitation state in other brain regions, we wish to argue that keeping the intensity constant might be the better choice for studies outside the motor cortex to avoid systematic differences between stimulation conditions. We used a similar procedure in our previous TMS studies (e.g. Kuhnke et al., 2017; Kuhnke et al.; 2020; Klaus & Hartwigsen, 2019; Meyer et al., 2018).

This has been clarified on page 15 as follows: “RMT was determined in the first session and held constant across both sessions to guarantee that the same intensity was used for both TMS sites and all conditions within each participant as in our previous studies (e.g. 25, 45, 56, 57).”

In summary, we think that separating the experiments in different sessions on different days while keeping the stimulation intensity constant is preferable for neurostimulation experiments, even if online protocols are used.

Comment 5. “For the stimulation protocol, was the site (pSTG, vertex) order counterbalanced across participants?” 

Reply 5. Yes, that was the case. This information has been added to page 12: “with the site order counterbalanced across participants”.

Comment 6. “Vocal responses: how many trials were rejected (>3ms) or discarded due to technical issues? Did the number of trials significantly differ among conditions? Please report.” 

Reply 6. This information has been added to page 16.

“All trials associated with these errors were coded as incorrect and excluded from the response time (RT) analyses, in addition to trials with RT > 3 s (1 trial in total). Trials with technical failures were discarded (at most 10 trials for each distractor by list by stimulation site combination). Discarded and incorrect trials were equally distributed across conditions (X-squared = 0.004, df = 6, p = 1).”

Comment 7. “TMS may induce blinking in participants thus interfering with viewing stimuli. Furthermore, acoustic stimulation (“clicking” sound of the coil) is known to provide a significant cross-modal resetting of occipital alpha oscillations (e.g., Romei and colleagues work). Since the authors were using visual stimuli, to what extend do they think these aspects could have influenced the observed patter of results?” 

Reply 7. This is an interesting point. We do not think that these issues might have influenced our results because we did not stimulate a visual (occipito-parietal) area. As for a potential “startle reflex”-like blinking, we experienced that participants usually habituate to the TMS sensation after having received a burst several times. Even if the clicking sound might have influenced cross-modal resetting of oscillations, this should have a similar impact on all task conditions. 

We now address this in the limitations section on page 25 as follows: “Finally, with respect to the TMS procedure, it should be noted that TMS may induce blinking in participants which may interfere with viewing stimuli or other side effects such as face twitches. Moreover, the acoustic stimulation has been demonstrated to induce cross-modal resetting of occipital alpha oscillations (e.g., 82). We are confident that these issues are unlikely to have strongly impacted our results. Even if the clicking sound might have influenced cross-modal resetting of oscillations or muscle twitching was present, this should have a similar impact on all task conditions and would thus be unlikely to explain our findings.” 

Discussion and interpretation of findings: 

Comment 8. “The authors mention that their findings are comparable/compatible with those observed in a study in the visual domain (Schwarzkopf, Silvanto & Rees, 2011), showing increased facilitation for more difficult targets. However, I don’t think this holds for the present study. It is true, that List 2 was somehow more difficult than List 1 but, in the overall context of the experiment, the congruent condition is still the easiest one, as shown by faster overall responses as compared to other conditions irrespectively of the TMS session.” 

Reply 8. Thank you for pointing this out. We agree with the reviewer that the observed facilitatory effect for the more difficult list in the overall easiest condition of congruent prime and picture might not be comparable with the Schwarzkopf et al. study. Following the reviewer’s suggestions, we have deleted the comparison with the Schwarzkopf study from the discussion. Instead, we have implemented her/his suggestion to interpret our findings in terms of differences in the amount of priming between conditions within the state-dependency framework (see next comment and reply). 

Comment 9. The authors argue that their findings can be interpreted in light of the state-dependent approach and I completely agree with them. Here, the authors used a task in which the distractor is supposed to prime the picture name. If low name agreement in the congruent condition translates into less congruency between picture name and distractor (and thus less priming), then it is possible that TMS might have facilitated the less active neural populations during the task (e.g., see Cattaneo et al. 2008 and Silvanto & Pascual-Leone, 2008 for a review). 

Reply 9. Thank you for bringing this up, this is indeed a very plausible explanation. We have integrated this into the discussion section on page 24 as follows: “In our design, the distractor word should prime the picture name and vice versa. We speculate that low name agreement in the congruent condition may translate into less congruency between picture name and distractor and thus less priming. Within the framework of state-dependency, it is possible that TMS might have facilitated the less active neural populations during the task (see 80, 81), which may explain the observed selective facilitation of the congruent condition in the somewhat more difficult list (based on a significant effect of name agreement only in list 2).” 

References

Boroojerdi B, Meister IG, Foltys H, Sparing R, Cohen LG, Topper R. (2002). Visual and motor cortex excitability: a transcranial magnetic stimulation study. Clin Neurophysiol 113, 1501-1504.

Abel S, Dressel K, Weiller C, Huber W. (2012). Enhancement and suppression in a lexical interference fMRI-paradigm. Brain and Behavior 2, 109-127.

de Zubicaray GI, Hansen S, McMahon KL. (2013). Differential processing of thematic and categorical conceptual relations in spoken word production. J Exp Psychol Gen 142, 131-142.

Hartwigsen G. (2015). The neurophysiology of language: Insights from non-invasive brain stimulation in the healthy human brain. Brain Lang 148, 81-94.

Klaus J, Hartwigsen G. (2019). Dissociating semantic and phonological contributions of the left inferior frontal gyrus to language production. Hum Brain Mapp 40, 3279-3287.

Kuhnke P, Beaupain MC, Cheung VKM, Weise K, Kiefer M, Hartwigsen G. (2020). Left posterior inferior parietal cortex causally supports the retrieval of action knowledge. Neuroimage 219, 117041.

Kuhnke P, Meyer L, Friederici AD, Hartwigsen G. (2017). Left posterior inferior frontal gyrus is causally involved in reordering during sentence processing. Neuroimage 148, 254-263.

Meyer L, Elsner A, Turker S, Kuhnke P, Hartwigsen G. (2018). Perturbation of left posterior prefrontal cortex modulates top-down processing in sentence comprehension. Neuroimage 181, 598-604.

Piai V, Roelofs A, Acheson DJ, Takashima A. (2013). Attention for speaking: domain-general control from the anterior cingulate cortex in spoken word production. Front Hum Neurosci 7, 832.

---

## [Decision Letter · Decision Letter 1]

12 Nov 2020

Effects of transcranial magnetic stimulation over the left posterior superior temporal gyrus on picture-word interference

PONE-D-20-24893R1

Dear Dr. Hartwigsen,

We’re pleased to inform you that your manuscript has been judged scientifically suitable for publication and will be formally accepted for publication once it meets all outstanding technical requirements.

Kind regards,

Nicola Molinaro, Ph.D.

Academic Editor

PLOS ONE

Reviewers' comments:

Reviewer's Responses to Questions

**Comments to the Author**

1. If the authors have adequately addressed your comments raised in a previous round of review and you feel that this manuscript is now acceptable for publication, you may indicate that here to bypass the “Comments to the Author” section, enter your conflict of interest statement in the “Confidential to Editor” section, and submit your "Accept" recommendation.

Reviewer #1: All comments have been addressed

Reviewer #2: All comments have been addressed

2. Is the manuscript technically sound, and do the data support the conclusions?

Reviewer #1: Yes

Reviewer #2: Yes

3. Has the statistical analysis been performed appropriately and rigorously? 

Reviewer #1: Yes

Reviewer #2: Yes

4. Have the authors made all data underlying the findings in their manuscript fully available?

Reviewer #1: Yes

Reviewer #2: Yes

5. Is the manuscript presented in an intelligible fashion and written in standard English?

Reviewer #1: Yes

Reviewer #2: Yes

6. Review Comments to the Author

Reviewer #1: (No Response)

Reviewer #2: Thanks for the opportunity to read a revised version of the manuscript. I am pleased with how the authors addressed my comments and have no major concerns.

7. PLOS authors have the option to publish the peer review history of their article (what does this mean?). If published, this will include your full peer review and any attached files.

Reviewer #1: No

Reviewer #2: No

---

## [Editor Report · Acceptance letter]

16 Nov 2020

PONE-D-20-24893R1 

Effects of transcranial magnetic stimulation over the left posterior superior temporal gyrus on picture-word interference 

Dear Dr. Hartwigsen:

I'm pleased to inform you that your manuscript has been deemed suitable for publication in PLOS ONE. Congratulations! Your manuscript is now with our production department. 

Kind regards, 

on behalf of

Dr. Nicola Molinaro 

Academic Editor

PLOS ONE